# Controlling Floquet states on ultrashort time scales

Matteo Lucchini [1,2] ✉, Fabio Medeghini[1], Yingxuan Wu[1,2], Federico Vismarra [1,2], Rocío Borrego-Varillas [2], Aurora Crego[2], Fabio Frassetto [3], Luca Poletto[3], Shunsuke A. Sato [4,5], Hannes Hübener [5], Umberto De Giovannini [5,6], Ángel Rubio [5,7] & Mauro Nisoli [1,2]

The advent of ultrafast laser science offers the unique opportunity to combine Floquet engineering with extreme time resolution, further pushing the optical control of matter into the petahertz domain. However, what is the shortest driving pulse for which Floquet states can be realised remains an unsolved matter, thus limiting the application of Floquet theory to pulses composed by many optical cycles. Here we ionized Ne atoms with few-femtosecond pulses of selected time duration and show that a Floquet state can be observed already with a driving field that lasts for only 10 cycles. For shorter pulses, down to 2 cycles, the finite lifetime of the driven state can still be explained using an analytical model based on Floquet theory. By demonstrating that the amplitude and number of Floquet-like sidebands in the photoelectron spectrum can be controlled not only with the driving laser pulse intensity and frequency, but also by its duration, our results add a new lever to the toolbox of Floquet engineering.

Due to its potential impact in many technological fields, the modification and control of materials properties with optical pulses has attracted a lot of attention from the scientific community[1–3]. Most notably, it led to the recent proposal of Floquet engineering, whose ultimate goal is to induce new properties and functionalities in driven materials that are absent in the equilibrium counterpart by using time-periodic external fields[4]. Light-induced superconductivity[5], Floquet topological states[6,7], Floquet phase transitions[8,9], anomalous Hall effects[10], and Spin-Floquet magneto-valleytronics[11] are but a few remarkable examples of phenomena that can be induced by periodic light driving[12]. All this becomes even more relevant when combined with ultrashort driving pulses[13,14], as it could provide us with the fascinating possibility to switch the physical properties of quantum materials on ultrafast time scales, establishing the ultimate switching limits for the next-generation devices. In graphene, for example, the

combination of Floquet engineering with ultrashort driving could be used to induce anomalous Hall effect[10], thus potentially giving unprecedented ultrafast control of a device current. In spite of its interest, it is not yet clear to what extend Floquet states can be created in a material, because a truly continuous driving pulse would damage the material. Hence, it is of paramount importance to clarify whether the Floquet formalism can be applied with short pulses and to what degree the effect of not perfectly periodic driving pulses can be described with the Floquet theorem[15]. Can a theory originally developed for continuous periodic driving field be extended to ultrashort pulses? Recent studies of the optical Stark effect in monolayer WS$_2$ suggest that Floquet theory should hold with pulses as short as ~15 optical cycles, but fail in identifying its limit, which should lie at even shorter pulse durations[16]. Therefore, the number of optical cycles needed for a Floquet state to be observed remains unclear. Note that,

[1]Department of Physics, Politecnico di Milano, 20133 Milano, Italy. [2]Institute for Photonics and Nanotechnologies, IFN-CNR, 20133 Milano, Italy. [3]Institute for Photonics and Nanotechnologies, IFN-CNR, 35131 Padova, Italy. [4]Center for Computational Sciences, University of Tsukuba, Tsukuba 305-8577, Japan. [5]Max Planck Institute for the Structure and Dynamics of Matter, 22761 Hamburg, Germany. [6]Università degli Studi di Palermo, Dipartimento di Fisica e Chimica-Emilio Segrè, I-90123 Palermo, Italy. [7]Center for Computational Quantum Physics (CCQ), The Flatiron Institute, New York, NY 10010, USA. ✉e-mail: matteo.lucchini@polimi.it

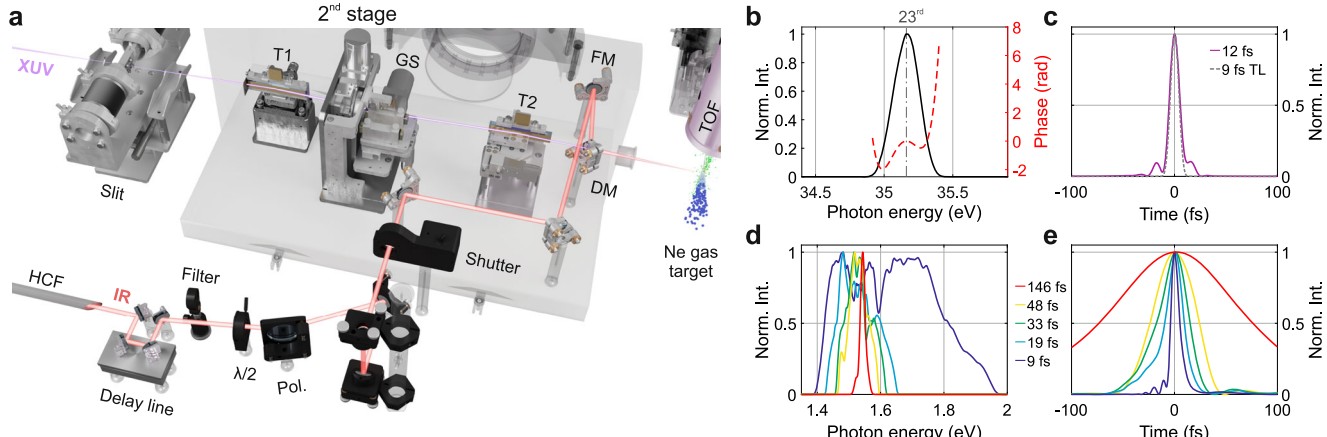

**Fig. 1 | Experimental setup and light pulses. a** Scheme of the experimental setup showing the second stage of the TDCM used to select the 23rd harmonic while preserving its time duration (T1 and T2 toroidal mirrors, GS grating, FM focusing mirror, Pol polarizer). The IR beam is compressed with a hollow-core fiber (HCF) setup and collinearly recombined to the XUV radiation through a drilled mirror (DM). Both beams are focused onto a Ne gas target, where the photoelectron spectra are collected by a time-of-flight (TOF) spectrometer. **b**, **c** Spectral and temporal properties of the XUV radiation used in the experiment. The spectral phase in **b** has been retrieved with a reconstruction algorithm (see Supplementary Section S1.2.1). **d**, **e** Spectral and temporal profiles of the IR pulses used in the experiment.

in addition to the creation of Floquet states, the stability of the Floquet states[17–19] and their population[20] are also important factors that affect the applicability of this concept for novel technologies.

In this work, we investigate the formation of a Floquet state of the simplest system capable of interacting with an external field: an electron in the continuum manifold of an atom—a quasi-free electron. This allowed us to find an experimental answer to the important questions listed above without losing generality. In particular, we ionized a Ne atom with a 12-fs extreme-ultraviolet (XUV) pulse while the system was dressed by few-fs infrared (IR) pulses of various time duration down to ~3.4 optical cycles. The IR dressing creates additional sidebands (SBs) in the photoelectron spectrum[21] that can be directly linked to the spectral components of the Floquet state (the so-called Floquet ladder). In combination with an analytical model, our results demonstrate that a Floquet-like approach can be still used to describe the light-induced state, if both the XUV and IR last for more than two cycles of the driving field. Furthermore, we found that the number of driving-field cycles necessary for the Floquet state to be observed depends both on the number of SBs involved (i.e., on the driving intensity) and on the time duration of the XUV pulse. In the short-pulse and low-intensity limit, our results prove that a Floquet state can be fully formed already within ten optical cycles. Proving the applicability of the Floquet formalism to ultrashort driving pulses, our work deepens the comprehension of fundamental light-induced phenomena and suggests a possible new pathway to realize metastable exotic quantum phases of matter and to exert control over their unique properties with unprecedented speed[22–24].

## Results

### Monochromatic driving

When a system is irradiated by a monochromatic field of frequency $\omega_0$, its Hamiltonian becomes periodic in time with a periodicity $T = 2\pi/\omega_0$. In analogy to the Bloch theorem for spatial crystals, the Floquet theorem predicts that the solution of the system Hamiltonian is given by the product of a time-periodic function and a characteristic phase factor[4,25]. $|\Phi_\gamma(t)\rangle = e^{-i\varepsilon_\gamma t}|\psi_\gamma(t)\rangle$. The index $\gamma$ runs over the states of the unperturbed system, the periodic function $|\psi_\gamma(t)\rangle = |\psi_\gamma(t+T)\rangle$ represents the Floquet state while $\varepsilon_\gamma$, defined up to an integer multiple of $\omega_0$, is called Floquet quasi-energy in analogy with the Block quasi-momentum $k$[26]. Each Floquet state can be developed in Fourier series to be written as a sum of Floquet ladder states of order $n$, amplitude $A_n$ and normalized spatial function $|\alpha(n,\gamma)\rangle$:[27] $|\psi_\gamma(t)\rangle = \sum_n A_n e^{in\omega_0 t}|\alpha_{n,\gamma}\rangle$.

The index $n$ can be interpreted as a position in the Floquet dimension, or equivalently, as the number of driving field photons involved[28]. Due to its generality, the Floquet theory finds application in a variety of scientific fields, from electronic systems to cold atoms[29] and photons in waveguide arrays[30]. Here we used a time-delay compensated monochromator[31] (TDCM, Fig. 1a) to generate short XUV pulses around 35.1-eV photon energy[32] (Fig. 1b, c) and ionize a Ne gas target. Few-femtosecond pulses centered around 800 nm and of controlled duration (Fig. 1d, e) dress the electron's final state. If the IR pulse is long enough to be considered monochromatic (red curve in Fig. 1d, e), it induces a particular Floquet state called Volkov state[21,33] (Fig. 2a), where the ladder state amplitudes and quasi-energy are given by (atomic units are used hereafter):

$$\begin{cases} \varepsilon = \frac{p^2}{2} + U_p \\ A_n(\mathbf{p}, \boldsymbol{E}_0, \omega_0) = \widetilde{J}_n\left(-\frac{\mathbf{p}\cdot\boldsymbol{E}_0}{\omega_0^2}, -\frac{U_p}{2\omega_0}\right) \end{cases} \quad (1)$$

$\mathbf{p}$ is the electron final momentum, $\boldsymbol{E}_0$ is the driving field amplitude, $U_p = E_0^2/4\omega_0^2$ is its ponderomotive energy and $\widetilde{J}_n$ indicates the generalized Bessel function of order $n$ (see Supplementary Section S2.2). Since the IR field can efficiently dress only the electron final state, here we consider a single Floquet state and drop the index $\gamma$ in the notation.

Under the strong-field (SFA) and dipole approximations[34], the photoelectron spectrum resulting from the ionization to the final dressed state $|\Phi(t)\rangle$ can be calculated as[34] $|\int_{-\infty}^{\infty}\langle\Phi(t)|\mu E_x(t)|0\rangle dt|^2$ where $|0\rangle = \phi_0(\mathbf{r})e^{iI_p t}$ is the atomic initial state of binding energy $I_p$, $E_x(t) = E_{x0}(t)e^{-i\omega_x t}$ is the XUV electric field and $\mu$ is the atomic electric dipole moment. Using the Floquet state defined by Eq. (1), it is possible to show that the resulting photoelectron spectrum is characterized by discrete SB peaks (Fig. 2b) with energy profile of the form (see Supplementary Section S2.4):

$$SB_n(\omega) = \left|\int_{-\infty}^{\infty} A_n(p, E_0, \omega_0)E_{x0}(t)e^{-i(\omega-\omega_n)t}dt\right|^2 = A_n^2(p_n, E_0, \omega_0)\left|\widetilde{E}_{x0}(\omega-\omega_n)\right|^2$$

$$(2)$$

where we have applied the central momentum approximation within the SB bandwidth to substitute $p \rightarrow p_n = \sqrt{2\omega_n}$ in the expression for $A_n$. The frequency $\omega_n = \omega_x + n\omega_0 - U_p - I_p$ is the central energy of the

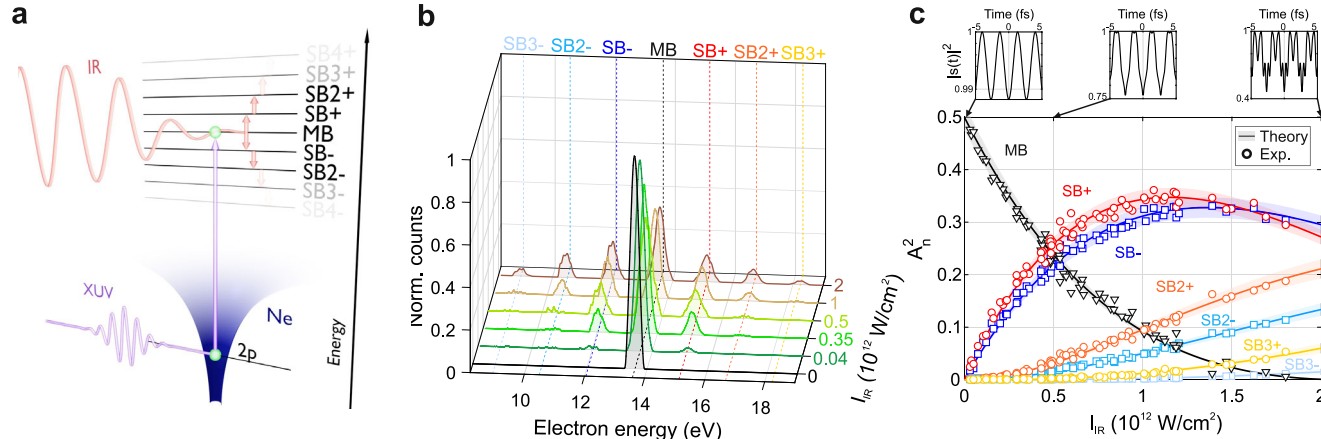

**Fig. 2 | Pulse intensity dependence. a** Cartoon of the experiment: an XUV femtosecond pulse ionizes the Ne atom to a state in the continuum, which is dressed by an IR pulse, inducing a time periodicity. In frequency, this corresponds to the creation of a Floquet ladder whose states, (called sidebands, SBs) are spaced by almost an IR photon. **b** Photoelectron spectra collected at selected IR intensities using the quasi-monochromatic pulse (-146 fs). The shaded areas represent the spectral area and are drawn to guide the eye. **c** Behavior of $A_n^2$ as a function of the IR intensity. The experimental values (markers) nicely follow the prediction of Eq. (1) (solid lines). The shaded areas represent the uncertainty over the calibration of the TOF transfer function. The main band signal (MB) is divided by 2. The upper panels show the Fourier transform in the time domain of the normalized scattering amplitude for $I_{IR}$ equal $10^9$, $5 \times 10^{11}$, and $2 \times 10^{12}$ W/cm$^2$.

SB of order $n$, $\widetilde{E}_{x0}(\omega)$ is the Fourier transform of the XUV pulse envelope and $\omega$ is the final photoelectron energy. Equation (2) suggests that a direct estimation of $A_n$ can be obtained by integrating $SB_n(\omega)$ in energy and dividing it by the area of the photoelectron spectrum obtained without the driving field (black curve in Fig. 2b), $I_0 = \int_{-\infty}^{\infty} |\widetilde{E}_{x0}(\omega)|^2 d\omega$. Figure 2c shows the value of $A_n^2$ extracted from the experimental photoelectron spectra with this procedure (open markers) while varying the IR field intensity, $I_{IR} = \frac{1}{2}\varepsilon_0 c E_0^2$, between $3 \times 10^{10}$ and $2 \times 10^{12}$ W/cm$^2$. In this intensity range we observe the formation of six SBs whose normalized amplitudes, $A_n$, nicely follows the prediction of Eq. (1) (solid curves with shaded area) and exhibit a non-monotonic behavior which depends on the Floquet order. Above ~$10^{12}$ W/cm$^2$, the amplitude of the Floquet ladder states with $n = \pm 1$ starts to decrease while the one of the higher order increases causing higher frequency components to become visible in the time evolution of the scattering amplitude related to the IR pump only, $|s(t)|^2$ (upper panels in Fig. 2c, see Supplementary Section S2.4). Therefore, this proves that by varying $I_{IR}$ it is possible to change relative weights of the spectral components of the final dressed state $|\psi(t)\rangle$.

## Pulsed driving

Once a direct link between the normalized amplitude of the photoelectron SB and the amplitudes of the Floquet ladder state has been established, it is natural to ask how the picture will change when the dressing light has a finite duration in time. In this case, the SB amplitude is expected to depend on the relative delay $\tau$ between the pulses[35]. Indeed, under the slowly-varying envelope approximation (SVEA) and for relatively low IR intensities (i.e., for $E_0(t) \rightarrow 0$), it is possible to show that the SB intensity becomes (see Supplementary Section S2.6):

$$SB_n(\omega, \tau) = \left| \int_{-\infty}^{\infty} \widetilde{J}_n\left(-p\frac{E_0}{\omega_0^2}, -\frac{U_p}{2\omega_0}\right) g(t)^{|n|} E_{0x}(t-\tau) e^{-i(\omega-\omega_n)t} dt \right|^2 \quad (3)$$

Equation (3) is formally identical to Eq. (2), suggesting that the photoelectron spectrum at each delay can still be thought as the squared modulus of the dipole matrix element between the initial state $|0\rangle$ and a Floquet-like final state $|\psi'(t)\rangle = \sum_{n \neq 0} A_n g(t)^{|n|} e^{in\omega_0 t} |\alpha_n\rangle + A_0'(t)|\alpha_0\rangle$, where each Floquet ladder state ($n \neq 0$) is the one generated by an equivalent

monochromatic driving field of amplitude $E_0$, multiplied by the $n$th positive power of the normalized pulse envelope, $g(t)$.

Figure 3a, b show the collection of photoelectron spectra as a function of $\tau$, obtained for $I_{IR} = 5 \times 10^{11}$ W/cm$^2$ and a full-width half-maximum (FWHM) duration of the IR pulse $\sigma_{IR} = 9$ and 45 fs, respectively. As expected, longer driving pulses generate an SB signal that lasts for a longer delay range[32]. In addition, even if the intensity $I_{IR}$ is kept constant, we found that longer pulses lead to more efficient SB population as indicated by the signal at $n = \pm 2$, visible only in Fig. 3b. To understand the origin of this behavior, we can follow the same procedure adopted for the monochromatic driving and estimate $A_n$ directly from the experimental traces. For Gaussian pulses, it is possible to show that this yields the following quantity (see Supplementary Section S2.7):

$$\Lambda_n^2 = \frac{A_n^2}{\sqrt{\frac{|n|\sigma_X^2}{\sigma_{IR}^2} + 1}} \quad (4)$$

where $\sigma_X$ is the FWHM time duration of the XUV pulse. Once the experimental spectra are corrected for the residual intensity fluctuations, the blue shift of the IR central wavelength and the deviations of the pulse envelopes from an ideal Gaussian shape (see Supplementary Section S2.2.1), $\Lambda_n^2$ can be directly extracted from the photoelectron traces and compared to the prediction of Eq. (4). Complete pump-probe scans are here performed not with the intent to follow the SB population in real-time during the laser-target interaction, but to precisely characterize the light pulses and locate the pump-probe temporal overlap where the signature of Floquet-like states is to be found. Open markers with error bars in Fig. 3c display the experimental results for $-2 \leq n \leq 2$, $\sigma_X \cong 12$ fs and an IR time duration variable between 9 and 146 fs (i.e., 3.4 and 55 cycles). The theoretical prediction is indicated by the solid curves. The shaded area represents the uncertainty associated with the TOF transfer function calibration (see Supplementary Section S1).

Also, in this case, the theoretical model nicely follows the experimental data, suggesting that the driving pulse duration can be used to control the temporal characteristic of the time evolution of the photoelectron wave packet associated solely with the IR dressing, $|s(t)|^2$ (upper panels in Fig. 3c).

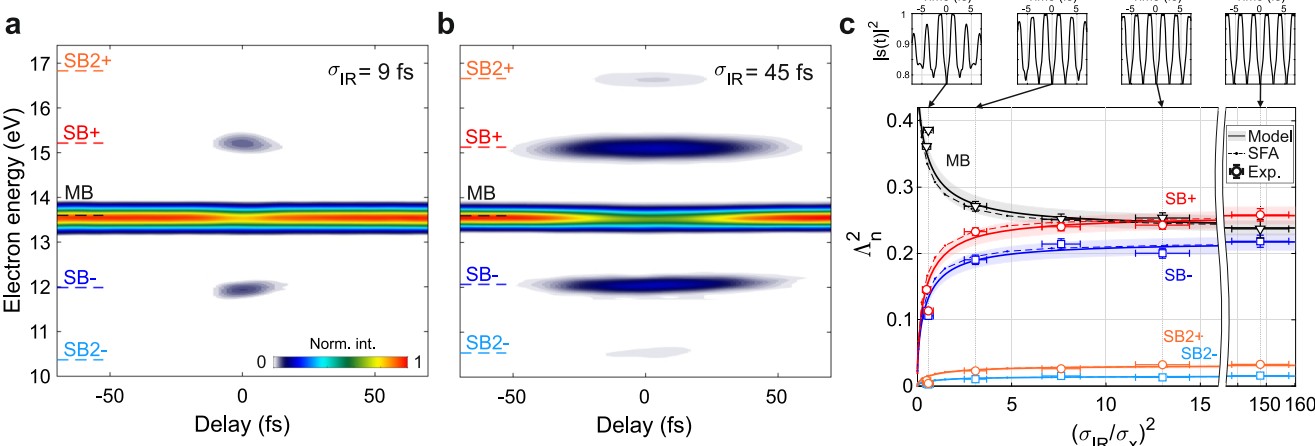

**Fig. 3 | Pulse duration dependence. a, b** Experimental spectrograms were obtained by ionizing Ne with a 12-fs XUV pulse and an IR pulse with a duration of 9 and 45 fs, respectively. **c** Behavior of the SB normalized intensities, $\Lambda_n^2$, as function of the ratio between the XUV and IR time durations. The experimental data (open markers) are obtained by changing the IR pulse duration while keeping its intensity fixed to ~ $5 \times 10^{11}$ W/cm². The solid line and shaded area represent the theoretical prediction of Eq. (4). The dashed curves with dots correspond to the same quantity extracted from the SFA calculations. The main band (MB) signal is divided by 2. The top panels show the Fourier transform in the time domain of the normalized scattering amplitude for $(\sigma_{IR}/\sigma_x)^2$ of about 0.6, 3.1, 13, and 148.

To test the validity of the approximations upon which the model of Eq. (4) is based, we simulated the experiment using only the SFA[36] (see Methods). The dash-dotted curves in Fig. 3c show the values of $\Lambda_n^2$ as extracted from the SFA simulations using the same procedure followed for the experimental data. The model (solid curves) nicely follows the SFA results (dash-dotted curves), deviating only for values of $\sigma_{IR} \simeq \sigma_X$. The main reason for the deviation can be traced back to the relatively high value of $I_{IR}$ used in the experiment to guarantee a reasonable signal-to-noise ratio and the fact that, for Eq. (4) to be accurate, the generalized Bessels, $\tilde{J}_n$, need to be monotonic in their argument (see Supplementary Section S2.6). In the momentum range under examination, this is true if $I_{IR} \lesssim 1 \times 10^{12}$ W/cm² where the deviation from the model stays below 10%. For intensities below $10^{11}$ W/cm², the agreement is instead almost perfect (provided that the pulses are long enough).

## Discussion

Once the theoretical model has been validated, Eq. (4) can be used to estimate the minimum number of driving cycles needed for a Floquet state $|\psi(t)\rangle$ to be observed as in the monochromatic case. Figures 4a and 4b show the simulated SB normalized intensity as a function of the IR and XUV number of cycles, respectively, for $I_{IR} = 1 \times 10^{10}$ W/cm². In Fig. 4a $\sigma_X = 11$ fs while in Fig. 4b $\sigma_{IR} = 10 T_{IR}$. Open dots represent the SFA simulations while the solid curves are the predictions of Eq. (4). The normalized curves for $n < 0$, not shown, are identical. For a pulse duration bigger than $2 T_{IR}$, the model correctly describes the system. Below two optical cycles, in Fig. 4a, the SVEA fails, while in Fig. 4b, the photoelectron trace is no longer characterized by discrete SB peaks as the XUV bandwidth becomes comparable to the IR photon energy. The non-trivial behavior of the SB amplitudes in Fig. 4b, where the XUV pulse is always shorter than the IR pulse, shows that the observed dependence cannot be solely attributed to a reduced temporal overlap between the IR and XUV pulses.

Using Eq. (4) we can now estimate how many IR cycles $N_{IR} = \sigma_{IR}/T_{IR}$ are needed for $\Lambda_n^2$ to reach its asymptotic value $A_n^2$. By expressing the XUV pulse duration in the number of IR cycles, $\sigma_X = N_X T_{IR}$, and calculating when the quantity $\frac{A_n^2 - \Lambda_n^2}{A_n^2}$ equals a threshold value $\alpha$, from Eq. (4) we obtain the asymptotic condition: $N_{IR} = \frac{(1-\alpha)\sqrt{|n|}}{\sqrt{2\alpha-\alpha^2}} N_X$ (see Methods). Therefore, the number of required driving cycles does not depend on the wavelength, but it is rather dictated by the maximum SB order, and ultimately by $I_{IR}$.

Figure 4c shows the results for $\alpha = e^{-3}$, corresponding to a $3\tau$-convergence; most notably, the number of required cycles increases with the SB order and with the time duration of the XUV pulse. If we consider the shortest XUV pulse for which clear SBs are observed, i.e., $N_X = 2$, and assume that $I_{IR} \leq 10^{11}$ W/cm² (so that SBs with $|n| \geq 3$ can be neglected), $3\tau$-convergence is reached after $8.5 T_{IR}$ for $n = \pm 2$ and $6 T_{IR}$ for $n = \pm 1$. This is different from the experimental condition used in Fig. 3, where the XUV pulse lasts almost 4.5 cycles. In that case $3\tau$-convergence is reached after $19.4 T_{IR}$ for $n = \pm 2$ and $13.7 T_{IR}$ for $n = \pm 1$. It is worth noting that the simulations of Fig. 4 are consistent with the time hierarchy discussed in ref. 14 since the condition $\Lambda_n^2/A_n^2 \to 1$ is met when $\sigma_{IR} > \sigma_X \gg T_{IR}$. Conversely from what suggested in the recent literature[14,37], our results show that Floquet-like bands can be observed, albeit with a reduced amplitude, also if the XUV (probe) pulse is longer than the IR (pump), $\sigma_X > \sigma_{IR}$, therefore proving that the Floquet theory can be extended to interpret the experimental results even if the hierarchy $\sigma_{IR} > \sigma_X \gg T_{IR}$ is not strictly matched. Finally, the explicit dependence on $|n|$ implies that the minimum number of pump cycles $N_{IR}$ scales differently for different SBs, with the important consequence that pulse duration alongside intensity can now be used to control the average population of the Floquet-like bands.

In summary, in this work, we used few-fs pulses of controlled duration and intensity to dress the photoelectron's final state. For quasi-monochromatic driving pulses, we demonstrated that there is a direct link between the observed SBs in the photoelectron spectrum and the amplitudes of the Floquet ladder states. With the help of an analytical model and numerical simulations, we studied the short-pulse limit, where we found that the final state can be interpreted in a Floquet-like picture as long as the two pulses are longer than two cycles of the driving light. Moreover, for short XUV pulses and moderate IR intensities, we found that only ~10 optical cycles are required for the final state to exhibit a Floquet ladder identical to the monochromatic case. Our results shed new light onto the formation of light-dressed states at the shortest time scales achievable with current technology, paving the way for the investigation and application of the ultimate speed limit of Floquet engineering. Furthermore, since our study proves that the Floquet-like ladder that is established by short pulses is strongly influenced by the temporal

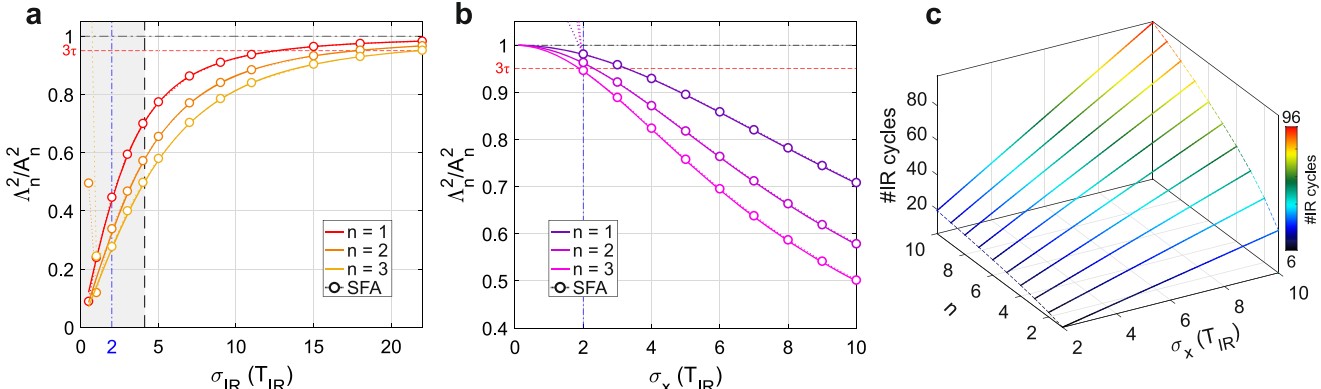

**Fig. 4 | Expected scaling with the number of cycles. a, b** Ratio between $\Lambda_n^2$ and its asymptotic value $A_n^2$ (monochromatic case), as a function of the IR and XUV pulse durations, while keeping fixed $\sigma_X = 11$ fs and $\sigma_{IR} = 10T_{IR}$, respectively. Open circles represent the SFA calculation, while the solid lines the model of Eq. (4). The higher the SB order n, the stronger the effect of the finite duration of the pulses. The blue dash-dotted vertical line sets the validity of the approximations used in the theoretical model. The black horizontal line corresponds to the monochromatic limit, while the red dashed line corresponds to the 3τ-convergence. The shaded gray area delimits the region where $\sigma_{IR} \leq \sigma_X$. **c** Number of IR cycles (also depicted by the false colors) necessary to reach 3τ-convergence as a function of the XUV time duration and the SB order. Floquet states composed of a higher number of SBs and generated by longer XUV pulses require longer times to establish.

profile of the exciting pulse, it suggests that the different timing of the excitation mechanism could be used to control the ratio between Floquet bands of bound or unbound states[7,19], thus allowing to disentangle the two channels in photoemission from solids.

## Data analysis

### Quasi-monochromatic pulses

After background removal, the SB amplitudes are extracted from the photoelectron spectra by integrating the SB signals in a 1-eV energy region around the corresponding peak and by normalizing them by the area of the XUV-only photoelectron spectrum. To correct for the not-constant transfer function of the TOF spectrometer (see Supplementary Section S1.1), the extracted values of each SB as a function of $I_{IR}$ are rescaled by a constant factor that minimizes the least-square distance between the curves. The uncertainty of this calibration factor is projected onto the theoretical curves and represented by the shaded area in Fig. 2c.

### Finite pulses

In this case, the experimental traces are recorded by changing the relative delay between the IR and XUV pulses with a step of 3–4 fs. Once the hollow-core fiber gas pressure has been set, the IR duration is measured with a second-harmonic FROG[38], and the pulse intensity is adjusted by rotating the $\lambda/2$-waveplate to obtain $I_{IR} \approx 5 \times 10^{11}$ W/cm². An XUV-only spectrum is taken for each five delay steps in order to correct for any deviation of the harmonic signal. The delay scan is repeated ten times for each IR duration and the final trace is obtained by averaging the individual scans. After background removal, the SB signals are integrated into a 1-eV energy window to evaluate their maximum value as a function of $\tau$. The theoretical curves of Fig. 3c have been calculated for an IR wavelength of 800 nm, $I_{IR} = 5 \times 10^{11}$ W/cm² and perfect Gaussian pulses. To account for the experimental deviations of these parameters, the photoelectron traces are reconstructed with an iterative algorithm (see Supplementary Section S1.2.1) to retrieve an accurate estimate of the exact pulse characteristics used in the experiment. Each experimental point is then corrected to account for the deviation of the experimental parameters for the ideal case to get the $\Lambda_n^2$ shown in Fig. 3c. Finally, the TOF transfer function is calibrated as done for the quasi-monochromatic measurements and its uncertainty is projected onto the theoretical predictions (shaded areas in Fig. 3c).

## Theoretical model and simulations

To check the validity of our model, we computed the photoelectron traces using the following SFA formula:[36]

$$S(\omega,\tau) = \left| \int_{-\infty}^{\infty} E_x(t-\tau) e^{-it\infty \int \frac{1}{2}(p+A_{IR}(t'))^2 dt'} e^{iI_p t} dt \right|^2 \quad (5)$$

where $A_{IR}$ is the IR vector potential defined by $E_{IR} = -dA_{IR}/dt$ and $\omega = p^2/2$. In the calculations, the IR and XUV fields are Gaussians: $E_0(t) = E_0 e^{-\frac{t^2}{\gamma_{IR}^2}}, E_{0x}(t) = E_{0x} e^{-\frac{t^2}{\gamma_X^2}}$. The complex quantities $\gamma_j$, with $j = X, IR$ for the XUV or IR pulse, are related to the intensity-FWHM of the pulses $\sigma_j$ and the group delay dispersion, $\beta_j$, by the following expression: $\gamma_j = 2\sqrt{\left(\frac{\sigma_j}{2\sqrt{2\ln(2)}}\right)^2 - i\frac{\beta_j}{2}}$.

For the case of monochromatic IR pulses, it is easy to show that Eq. (5) yields the same result as Eq. (2). Indeed, starting from the definition of $|0\rangle$ and $|\Phi(t)\rangle$ given in the text, and assuming $d_n \cong 1$ so that the dependence on the spatial part of the functions can be neglected, we can write:

$$\left| \int_{-\infty}^{\infty} dt \langle \Phi(t) | \mu E_x(t) | 0 \rangle \right|^2 \cong \left| \int_{-\infty}^{\infty} E_{0x}(t) \sum_n A_n e^{i\left[\frac{p^2}{2} - (\omega_x + n\omega_0 - U_p - I_p)\right]t} dt \right|^2 \quad (6)$$

which is identical to Eq. (5) if the IR field is written as $E_{IR} = \bar{E}_0 \sin(\omega_0 t)$ and the semi-classical action is expanded using the Jacoby-Anger formula[34] (see Supplementary Section S2.3). If the XUV spectrum is narrow enough, the modulus of the integral in Eq. (6) is identical to the integral of the modulus, hence Eq. (2) derives directly from the above equation.

In the case of finite pulse duration, starting from Eq. (5), applying the SVEA and using the following asymptotic limit for the generalized Bessel function of order $n \neq 0$, $\tilde{J}_n\left(-p\frac{E_0(t)}{\omega_0^2}, -\frac{U_p(t)}{2\omega_0}\right) \cong \tilde{J}_n\left(-p\frac{E_0}{\omega_0^2}, -\frac{U_p}{2\omega_0}\right)g(t)^{|n|}$ (see Supplementary Section S2.6), it is possible to obtain Eq. (3) from which Eq. (4) can be analytically derived for the case of Gaussian pulses[35] (see Supplementary Section S2.7). The number of IR cycles needed to reach 3τ-convergence can be

calculated from Eq. (4) by evaluating $\frac{A_n^2 - \Lambda_n^2}{A_n^2} = \alpha = e^{-3}$. Substituting the expressions for $A_n$ and $\Lambda_n$ one gets:

$$1 - \frac{1}{\sqrt{|n|\left(\frac{N_X}{N_{IR}}\right)^2 + 1}} = \alpha,$$ which can be inverted to obtain the expression of $N_{IR}$ reported in the main text.

## Methods

### Experimental setup

IR pulses with a time duration of about 35–40 fs, a center wavelength of 811 nm, a repetition rate of 1 kHz, and energy of ~0.8 mJ are focused onto a static gas cell filled with Ar to generate high-order harmonics. The 23rd harmonic is selected with a TDCM which works in a subtractive configuration, maintaining the XUV pulse time duration[32]. A portion of the IR beam (about 1 mJ), removed by a beam splitter prior to harmonic generation, is sent to a hollow-core fiber compression setup filled with Ne, where the pulse duration is controlled by changing the pressure of the filler gas. After the fiber, a $\lambda/2$-waveplate is used in combination with a polarizer to adjust the pulse energy without altering its time duration or its focal properties. A mechanical shutter is used to switch the IR radiation on and off during the experiments to collect the reference XUV-only signal. The IR beam is then recombined with the 23rd harmonic through a drilled mirror after passing through a delay stage equipped with a piezo controller. Both beams are focused onto a Ne gas target (Fig. 1a). The resulting photoelectron spectra are recorded with a time-of-flight (TOF) spectrometer, while an XUV spectrometer allows the inspection of the XUV spectral content at the end of the beamline. To obtain a quasi-monochromatic pulse, the IR beam is instead filtered by an interferential filter with 10-nm bandwidth.

## Data availability

The data generated and analysed in this study are provided in the Supplementary Information/Source Data file. Extended data were available from the corresponding author upon reasonable request. Source data are provided with this paper.

## Code availability

No custom codes have been used. The analytical formulas used in this study are reported in the main text and in the supplementary.

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

## Acknowledgements

This project has received funding from the European Research Council (ERC) under the European Union's Horizon 2020 research and innovation program (grant agreement No. 848411 title AuDACE). M.L. and L.P. further acknowledge funding from MIUR PRIN aSTAR, Grant No. 2017RKWTMY. A.G. and R.B.-V. received funding from the DINAMO project funded by Fondazione Cariplo (grant no. 2020-4380). S.A.S., H.H., U.D.G., and A.R. were supported by the European Research Council (ERC-2015-AdG-694097) and Grupos Consolidados UPV/EHU (IT1249-19).

## Author contributions

M.L. conceived the experiment, developed the theoretical model, and performed the calculations. F.M., Y.W., F.V., and A.C. performed the measurements. F.M. and M.L. evaluated and analysed the results. F.M. performed the photoelectron trace reconstructions. R.B.-V. and M.N. participated in the scientific discussion, data interpretation, and contributed to the definition of the experimental procedures. F.F. and L.P. designed and built the monochromator. S.A.S., U.D.G., H.H., and A.R. helped in the theoretical discussion of the data. M.L. wrote the first version of the manuscript to which all authors contributed.

## Competing interests

The authors declare no competing interests
