## [Peer Review File · Nature Communications]

Controlling Floquet states on ultrashort time scalesREVIEWER COMMENTS

Reviewer #1 (Remarks to the Author):

This manuscript is clear, well written and with an extraordinary scientific interest for a broad metrological community, although the technique is possible in the Politecnico di Milano laboratories and very few others in the world.

I recommend acceptance of the paper on its present form.

Reviewer #2 (Remarks to the Author):

The authors aim to identify when Floquet theory breaks down due to having short pulses, which is an important question and worth pursuing. However, the results presented in this manuscript do not provide a convincing test of these limits, and there are several questions around the analysis and interpretation.

Firstly, the authors claim that their experiments are driving Floquet states on the free electrons created by an XUV pulse. These photoelectrons will remain in the interaction region for $>ps$ after the XUV (ionization pulse), and thus the effect of dressing the free electrons (ie driving Floquet states) should be evident for a large range of delays with the IR pulse coming after the XUV pulse. Fig.3 a and b show that the sidebands are only evident when the two pulses are overlapped in time. It seems then that it is unlikely that the response observed is due to dressing free electrons.

What seems more likely is that the sidebands are generated "by the absorption of a harmonic photon and the additional absorption or emission of an IR photon" (or 2 IR photons in the case of the 2nd sideband). This is indeed the interpretation in REF 31, which involves many of the same authors of this manuscript, and shows very similar data (eg Fig 1, Fig 2a, and Fig 4).

The discussion of the effects of pulse duration appear to miss the important point. Specifically, I would posit that the changes to the sideband amplitudes as a function of pulse duration can simply be determined by the number of IR photons present through the XUV pulse duration (ie it should be proportional to $|\int_{-\infty}^{\infty} [g(t)]^n E(t) dt|^2$). In the case where the IR pulse is longer than the XUV pulse (and the peak intensity kept constant) the sideband amplitudes should be \sim constant, as observed. With the IR pulse shorter than the XUV pulse the sideband amplitude should decrease, as observed. In actuality, the equation (3) in the manuscript actually does have this form, however the interpretation and discussion appears to miss the point. The authors appear to suggest that this decrease in amplitude of the sidebands is due to the Floquet formalism breaking down, however the key origin of the decrease is simply due to the reduced amplitude of the overlap integral. With this simpler understanding of the results, it is apparent that even if the sidebands were present as a result of Floquet processes, there is no clear evidence of the Floquet formalism breaking down.

The authors suggestion that the pulse duration can be used as a means to control Floquet sidebands is also misplaced. The effect measured is an integrated response; thus increasing the pulse duration, with constant peak intensity, simply increases the pulse area (or total photon number) and integrated response.

There are several other issues, including: making conclusions based on fitting effectively 4 data points to the theory curve; the varying choice of ways to describe the pulse duration and clarity around these; and overall clarity of the manuscript. However, these are secondary to the more fundamental issues regarding interpretation outlined above.

Manuscript ID: NCOMMS-22-18068-T Type: research article

Title: Controlling Floquet states on ultrashort time scales

Authors: M. Lucchini, et al.

List of changes:

- We changed few sentences in the text to stress that the electron is in the proximity of the atomic potential and to avoid any possible misinterpretation regarding the major point made by Reviewer #2.
- Ref. 17 has been added to the manuscript in order to reply to Reviewer #2.
- Ref. 3 has been updated as the related work has now been published.
- We corrected few typos in the text.
- We added a data availability statement and a code availability statement.

We are grateful to all Referees for their work and useful comments. We are confident that the new version of the manuscript fully addresses their concerns and meets the criteria for publication on Nature Communications.

In the following, we give our reply point by point.

Reviewer #1

This manuscript is clear, well written and with an extraordinary scientific interest for a broad metrological community, although the technique is possible in the Politecnico di Milano laboratories and very few others in the world.

I recommend acceptance of the paper on its present form.

We thank the Reviewer for the positive comment and for underling the relevance of our work.

Reviewer #2

The authors aim to identify when Floquet theory breaks down due to having short pulses, which is an important question and worth pursuing. However, the results presented in this manuscript do not provide a convincing test of these limits, and there are several questions around the analysis and interpretation.

We thank the Reviewer for recognizing the importance of the treated topic. We are confident that the new version of the manuscript and our point-by-point reply address his/her main concerns and prove that we indeed investigate the limit of the Floquet theory.

Firstly, the authors claim that their experiments are driving Floquet states on the free electrons created by an XUV pulse.

To avoid any potential misunderstanding let us stress here that we do not drive a completely free electron with the IR pulse. This would be impossible as a completely free electron cannot absorb a photon or the momentum would not be conserved. We rather dress the final electron state in the continuum with the IR field and populate this dressed final state with the XUV pulse. This difference is subtle but a key point as it will become clear in the following discussion.

These photoelectrons will remain in the interaction region for >ps after the XUV (ionization pulse), and thus the effect of dressing the free electrons (ie driving Floquet states) should be evident for a large range of delays with the IR pulse coming after the XUV pulse. Fig.3 a and b show that the sidebands are only evident when the two pulses are overlapped in time. It seems then that it is unlikely that the response observed is due to dressing free electrons.

While it is true that the photoelectron lives for long times, a totally free electron cannot absorb a photon (otherwise the total momentum would not be conserved). Photon absorption/emission occurs only when the electron is still interacting with its nucleus such that the nucleus can assure momentum conservation. This is clearly stated in the paper from L. B. Madsen (Ref. 34 of the current manuscript): *“The laser light is assumed to be linearly polarized, and the dipole approximation ... may be applied very accurately, where the relevant values of \mathbf{r} are determined by the fact that exchange of photons takes place only within the range of the atomic potential, $\sim a_0$.”*

Therefore, sideband (SB) formation happens only during the ionization process and it is limited to the region of temporal overlap between the XUV and IR pulses. It is not possible to induce the formation of a SB once the electron is totally free and too far away from its nucleus.

While the widely used photon picture is very intuitive, we believe that it is better to look at the problem from another (complementary) perspective. As we are dealing with a strong IR field, it is better to include its vector potential into the potential of the system Hamiltonian (as done in the Floquet formalism). This is the same approach followed by L. B Madsen in his very didactical paper (see Ref. 34 of the current manuscript).

Let's start by assuming a monochromatic IR pulse. As reported in Ref. 33, the solution to the time-dependent Schrödinger equation, which includes the coupling with the IR field, is a Volkov wave (eq. (6) in Ref. 34). This wavefunction represents a particular Floquet state with quasi energy and amplitudes given by eq. (1) of the manuscript

$$\left\{ \begin{array}{l} \varepsilon = \frac{p^2}{2} + U_p \\ A_n(\mathbf{p}, \mathbf{E}_0, \omega_0) = \tilde{J}_n\left(-\frac{\mathbf{p} \cdot \mathbf{E}_0}{\omega_0^2}, -\frac{U_p}{2\omega_0}\right). \end{array} \right.$$

In section IV, “Laser-assisted photoelectric effect”, Madsen describes the photoemission initiated by an energetic photon when an atom is dressed by a strong laser field. This process corresponds to our experiment with long IR pulses (called quasi-monochromatic in our case). In his paper Madsen writes:

“The Born approximation for the S-matrix expression for a transition from the initial atomic wave function Φ_0 to a final Volkov wave function ψ_V is

$$(S - 1)_{fi}^\beta = -i \int_{-\infty}^{\infty} dt \langle \psi_V | \mathbf{A}_V(t) \cdot \hat{\mathbf{p}} | \Phi_0 \rangle \quad (31)$$

where $\Phi_0 = \phi_0(\mathbf{r})e^{iE_b t}$ with E_b the binding energy of the initial electron wave function. Here $\mathbf{A}_V(t)$ is the vector potential of the high-energy photon.”

Within the dipole approximation, this equation corresponds to our eq. (6). It predicts the formation of sidebands spaced by an IR photon in the photoelectron spectrum and shows that the final state for

photoemission is indeed a Floquet (Volkov) state. The validity of this model for monochromatic fields (or relatively long pulses) has been proven in many experiments, one of which is reported in figure 2c of our manuscript. Another remarkable example can be found in T.E. Glover et al., Phys. Rev. Lett **76**, 2468 (1996), now Ref. 17 in the manuscript. In this work, the ionization is done by a comb of harmonics. The SBs that originate with the target atoms dressed by an IR field are well reproduced by projecting the final electron state into Volkov waves. Indeed, the authors write: “*Measured free-free scattering amplitudes are well described by approximating the final electronic wave function as that of free particle oscillating in the laser field, i.e., as a Volkov wave function.*”. The authors also state: “*The most significant deficiency in the present theory is the neglect of modifications to the final electronic wave function resulting from the presence of the Coulomb potential. The agreement observed between data and theory indicates that these effects are not severe, a result consistent with the expectation that atomic “dressing” of the Volkov wave is most significant near the (x-ray) photoionization threshold.*”

In view of this, we are confident in stating that the SBs observed with (quasi-)monochromatic pulses arise from a transition to a Floquet final state.

In case of IR pulses of finite duration, if the XUV ionizes the atom too early or too late, the IR amplitude is zero, the state in the continuum is not dressed and we do not observe SBs. If ionization by the XUV photons happens while the external IR field is non-zero (i.e., within the time overlap region), the final state for the electron is a solution of the Hamiltonian that includes the IR vector potential, i.e. a dressed Floquet-like state, and SBs are observed.

Our results and theoretical derivation show that, for the case of finite pulses, if the IR envelope is opportunely considered, the electron final state can be treated with the same model used for the monochromatic driving and based on Floquet theory. Therefore, our work indeed addresses the validity of the Floquet theory and the possibility of its non-adiabatic extension to the finite driving limit. This constitutes a non-trivial result, hard to investigate experimentally and, to our knowledge, never observed/studied before.

To avoid any possible form of misinterpretation, we have carefully revised the manuscript and changed the text so that it is now clear that the IR dresses the final state and this can be well approximated with a Volkov wave, exact solution for the free electron. For example, at page 3 we changed “dress the free electron” with “dress the electron final state”. Later we changed “Since the IR field can efficiently dress only the free-electron state” with “Since the IR field can efficiently dress only the electron final state”. In the conclusions we changed “In summary, in this work we used few-fs pulses of controlled duration and intensity to induce a Floquet state of a free electron.” with “In summary, in this work we used few-fs pulses of controlled duration and intensity to dress the photoelectron final state.”

What seems more likely is that the sidebands are generated “by the absorption of a harmonic photon and the additional absorption or emission of an IR photon” (or 2 IR photons in the case of the 2nd sideband). This is indeed the interpretation in REF 31, which involves many of the same authors of this manuscript, and shows very similar data (eg Fig 1, Fig 2a, and Fig 4).

We reply to this point with a double comment:

1) The fact that IR photons are absorbed or emitted is not in contrast with the photoelectron final state being a Floquet-like state. This is commonly accepted. For example, while discussing laser-assisted photoemission, in his paper, L. B. Madsen writes: “*The sidebands correspond to the exchange of photons with the laser field*”. Nevertheless, the photoelectron final state is a Volkov state as we discussed in the previous point. The quantum picture (photon exchange) and the classical picture (classical IR vector potential included in the system Hamiltonian) do not exclude one another, but rather coincide at their limit.

2) The scope and the results reported in Ref. 31 (now Ref. 32) have nothing to do with what reported in the present manuscript. In Ref. 31 (now 32) we investigated the possibility to retrieve the temporal properties of IR and XUV pulses from the pump-probe trace with Ptychographic algorithms. For this reason, only the delay-energy distribution of the first sideband is investigated. Even if two experimental spectrograms are reported in Fig. 5 of Ref. 31 (now 32), the strength of the sideband signal, its exact dependence on the IR intensity and its scaling with the pulses time duration are not discussed. The interpretation in terms of Floquet states and the data and analysis we report in the current manuscript are new and not contained/discussed in Ref. 31 (now 32).

The discussion of the effects of pulse duration appear to miss the important point. Specifically, I would posit that the changes to the sideband amplitudes as a function of pulse duration can simply be determined by the number of IR photons present through the XUV pulse duration (ie it should be proportional to $|\int_{-\infty}^{\infty} [g(t)]^n E(t) dt|^2$). In the case where the IR pulse is longer than the XUV pulse (and the peak intensity kept constant) the sideband amplitudes should be \sim constant, as observed. With the IR pulse shorter than the XUV pulse the sideband amplitude should decrease, as observed. In actuality, the equation (3) in the manuscript actually does have this form, however the interpretation and discussion appears to miss the point. The authors appear to suggest that this decrease in amplitude of the sidebands is due to the Floquet formalism breaking down, however the key origin of the decrease is simply due to the reduced amplitude of the overlap integral. With this simpler understanding of the results, it is apparent that even if the sidebands were present as a result of Floquet processes, there is no clear evidence of the Floquet formalism breaking down.

We would like to stress that if everything is done correctly, there should not be contradiction between the quantum (photon) interpretation and the classical (Floquet) description. At the limit for a big number of photons, the two models should lead to the same result (see F. H. M. Faisal, *Theory of Multiphoton Processes* (Springer US, 1987).).

This said, the equation the Referee mentions, eq. (3) of the manuscript, shows that the SB signal is proportional to the Fourier transform of the product between $g(t)^{|n|}$ and the XUV envelope $E_{0x}(t)$, which does not give the same result as the formula suggested by the Reviewer: $|\int_{-\infty}^{\infty} g(t)^n E(t) dt|^2$. Figure R1(a) reports the ratio between the monochromatic SB signal and the finite-pulse case as calculated with our model (red scale, same data as in Fig. 4a of the manuscript) and as calculated with the equation reported by the Reviewer (gray scale). In this last case a monochromatic field, i.e. $g(t)^{|n|} = 1$, would lead a signal proportional to $|\int_{-\infty}^{\infty} E(t) dt|^2$. So, Λ_n^2/A_n^2 in the Reviewer's model becomes:

$$\frac{|\int_{-\infty}^{\infty} g(t)^n E(t) dt|^2}{|\int_{-\infty}^{\infty} E(t) dt|^2}$$

The value of this quantity for $n = 1, 2$ and 3 , and XUV duration of 11 fs is reported in black, dark grey and light grey. As it is possible to notice the model suggested by the Reviewer underestimates the SB intensity for a given IR duration, independently of the sideband order. Moreover, the model suggested by the Reviewer is slower in reaching its asymptote and predicts a change of curvature for short pulse durations, not observed in our model or in the SFA calculations (see Fig. R1(b) and R1(c)).

Figure R1: **(a)** Ratio between Λ_n^2 and its asymptotic value A_n^2 as calculated by our Floquet-like model (red-scale curves) and the model proposed by the Reviewer (grey-scale curves). The parameters used for the calculations are the same as Fig. 4a of the main manuscript. The red curve should compare to the black, the orange to the dark grey and the yellow to the light grey. **(b)** Comparison between our model and the SFA simulations (same data as in Fig. 4a of the main manuscript). **(c)** Comparison between the SFA simulations and the model proposed by the Reviewer. As it is possible to observe the latter does not reproduce the simulation results.

The authors suggestion that the pulse duration can be used as a means to control Floquet sidebands is also misplaced. The effect measured is an integrated response; thus increasing the pulse duration, with constant peak intensity, simply increases the pulse area (or total photon number) and integrated response.

While we do not question that by changing the pulse duration with fixed intensity also the number of photons changes, our experimental results clearly show that the sideband intensity depends on the pulse duration. Therefore, independently from the exact microscopic mechanism, it is undoubted that the Floquet sideband intensity can be changed by controlling the dressing pulse duration.

Let us now investigate the dependence on the number of photons. For a pulse with Gaussian envelope, the area is given by:

$$Area = \int_{-\infty}^{\infty} A_0^2 e^{-\frac{2t^2}{\gamma^2}} dt = A_0^2 \sqrt{\frac{\pi}{2}} \gamma = A_0^2 \sqrt{\frac{\pi}{2}} \frac{FWHM}{\sqrt{2 \log(2)}},$$

where FWHM is the full-width half-maximum time duration of the pulse and the pulse peak intensity is given by $I = \frac{1}{2} \epsilon_0 c A_0^2$ being A_0 the field amplitude. The area, i.e. the number of photons, is thus directly proportional both to the pulse intensity and duration. To prove that the observed dependence does not simply depend on the number of IR photons we calculated the expected sideband strength as a function of the IR pulse area for two cases: (i) the field intensity is kept fixed at 5×10^{11} W/cm² while the IR time duration is

varied; (ii) the IR time duration is kept fixed at 50 fs while the IR field intensity is varied. The results for sideband $n = 1$ and $n = 2$ are reported in Fig. R2(a) and Fig. R2(b), respectively. As it is possible to notice, even if the area, i.e. the number of photons, is changed in the same range, the sideband intensity shows a different behaviour.

The reason is the following: in the intensity region where our model is valid, i.e. below $\sim 1 \times 10^{12}$ W/cm², equation (S46) of the supplementary material can be used. The normalized sideband intensity is then given by eq. (S58) (equivalently, eq. (4) in the main manuscript) where the field amplitude appears as the argument of the Bessel function while the IR pulse duration appears within the square root at the denominator.

Therefore, changing the total number of photons by acting on the pulse intensity or on the pulse duration does not have the same effect. This allows us to conclude that what we observe is not simply related to the number of photons in the pulse.

Figure R2: **(a)** Normalized sideband intensity for $n = 1$ (red curve) and $n = 2$ (orange curve) as a function of the IR pulse area for a fixed pulse intensity of 5×10^{11} W/cm². The vertical dashed blue line marks the IR pulse duration used in (b). **(b)** Normalized sideband intensity for $n = 1$ (red curve) and $n = 2$ (orange curve) as a function of the IR pulse area for a pulse duration of 50 fs. The vertical dashed blue line marks the IR intensity used in (a).

There are several other issues, including: making conclusions based on fitting effectively 4 data points to the theory curve;

If the Reviewer refers to the data in Figure 3c, the data points are 6 (the shortest duration has two points) and we did not perform a fit. The model results (solid curves) are plotted against the experimental data. We would like to stress that these measurements are experimentally not easy. As the sideband signal depends also on the exact IR wavelength and intensity, it is necessary to obtain a fine control of all these parameters to be able to obtain meaningful data. With our current setup, already at the state of the art as the Review #1 pointed out, it is not possible to scan the time duration with a finer step while keeping the requested level of control over the other parameters. Nevertheless, we would like to stress that our main finding, that the sideband amplitude decreases with decreasing IR duration, is a generally observed trend that is not derived from a fit and is robust and clear already in the presented data.

the varying choice of ways to describe the pulse duration and clarity around these;

The time duration of the pulses is defined as the full-width half-maximum (FWHM) of the pulse intensity profile both for the experimental data and for the computations. For the calculations, the FWHM duration is defined at the end of section S2.1 of the supplementary material.

We are aware that there are other possible definitions for the pulse durations. As long as the same definition is consistently used, the following conclusions do not depend on it. For this reason we decided to use the FWHM duration throughout the whole work.

and overall clarity of the manuscript. However, these are secondary to the more fundamental issues regarding interpretation outlined above.

We are confident that the revised manuscript is clearer and that after our point-by-point reply the Reviewer can agree with our interpretation.

REVIEWER COMMENTS

Reviewer #3

The authors have provided a detailed response and added some clarification which has greatly helped. In my opinion, the experiments and analyses are excellent (notwithstanding some concerns in the specific comment #1 below), however, I find their discussion of some of the implications misplaced.

The authors have clarified that the states being dressed by the IR laser pulse are free-electrons. As clarified, they are not free electrons, but the final state of the electrons in the continuum of states, still associated with the nucleus. Thus they are only present while the XUV laser pulse is present. This clarifies one of the previous confusions.

This is, however, a very specific case of Floquet states, and in contrast to what is stated on line 53-54, I believe it does lose generality (and is not looking at “the simplest system: a free electron”). Many of the results and conclusion arise due to the interplay between the XUV and IR pulse durations. To be able to make general statements on the applicability of Floquet formalism for short pulses requires the effects of the XUV pulse duration to be taken into account.

The authors do in fact do this through their detailed model and conclude that for driving pulses down to 2 cycles the results “can still be explained using an analytical model based on Floquet theory”. In other words, the reduction seen in the amplitude of the sidebands for short IR pulses is due to a combination of the IR and XUV pulse durations, and the Floquet formalism still holds for driving pulses as short as 10 optical cycles. This is an important point and valuable observation.

Where I disagree is with the assertion that under the specific conditions “*~10 optical cycles are required to establish a Floquet ladder identical to the monochromatic case.*” Certainly, under those conditions 10 optical cycles are required to generate side bands with the same amplitude as the monochromatic case, however, this is not the same as the time to establish the Floquet ladder, or as they claim, the “*minimum number of driving cycles needed for a Floquet state $|\psi(t)\rangle$ to be established*”.

The measured sideband amplitude is time-integrated, thus when the IR pulses become similar to or shorter than the XUV pulse, these amplitudes do not represent the instantaneous population or presence of the Floquet states. The asymptotic condition (Line 213) therefore does not represent the time to establish or populate the Floquet states. All of this is taken into account in the authors’ detailed and thorough model, which is followed down to 2 optical cycles and suggests that the instantaneous population of the Floquet states are established at least this fast. That is, they do not need 10 optical cycles to be established.

The authors’ detailed response in their rebuttal letter does not preclude these points. I agree with their detailed analysis, but again, not their conclusions. Fig R1 compares two different things (one the peak sideband amplitude, one effectively the integrated amplitude) and I do not disagree that their model is more complete than a simple phenomenological picture. The point remains, however, that the reduction in sideband amplitude is primarily due to the pulse overlap, and not any measured delay in formation or population of the Floquet states. If this is not the case, then I would welcome clarification, but as it stands I do not see any evidence of this.

In Fig R2, I wouldn’t expect varying the intensity and the pulse duration to give the same result, primarily because the XUV pulse duration relative to the IR pulse duration is also important.

I agree with their point that “*the Floquet sideband intensity can be changed by controlling the dressing pulse duration.*” However, what is claimed in the manuscript goes beyond just discussing the sideband intensity. As discussed above, the sideband intensity is a time-integrated measurements and is not the same as controlling the instantaneous population or presence of the

Floquet states. Thus I continue to disagree with the claim that “*the Floquet ladder population is strongly influenced also by the temporal profile of the exciting pulse*”. Again, however, if the authors are able to show evidence of a deviation of the instantaneous population of Floquet states from the Floquet formalism (incorporating the finite XUV and IR pulse durations) then this claim would be more reasonable.

Further specific comments:

1. For the data in Fig 3c, *the experimental spectra are corrected for the residual intensity fluctuations, the blue shift of the IR central wavelength and the deviations of the pulse envelopes from an ideal Gaussian shape (see Supplementary Section S1.2.1),*
 The raw data are shown in FigS2 and show significantly more scatter. It seems from my understanding that the process for correcting these relies on fitting the delay dependent photoelectron spectra. One of the largest corrections seems to be to the IR pulse intensity, which varies by over a factor of 2. It is not clear to me why this is necessary when the pulse properties are determined by FROG and the intensity is measured based on these. What is the reason for these differences? What is perhaps more concerning is that in the fitting process, the sidebands are used and there is a term in the equation describing them goes as the electric field amplitude of the XUV beam multiplied by the amplitude of the IR beam, just as there is in Eq4 of this manuscript. While I haven't fully delved into the fitting procedure, I am a little worried that there may be a circular argument here, where the expected response is used to determine the intensity (ie the fitting procedure detailed in S1.2 and Ref S35), which is then used to shift the experimental data points in Fig 3c, which then agree with the model (Eq4) and is used as evidence that the model is accurate and followed by the data. I acknowledge that the models are different, but there are certainly similarities, and it would aid clarity if the authors can give more detail on this fitting. It would also help if they give some explanation as to why the IR intensity varies by more than a factor of two from what was expected.
2. The authors state: *Most interestingly, our results show that Floquet-like bands can be observed, albeit with a reduced amplitude, also if the XUV (probe) pulse is longer than the IR (pump), $\sigma \gg \tau$ therefore extending the previously known regime of applicability of Floquet theory.*
 Floquet theory is primarily used to describe dressing of equilibrium states, which equivalent to a regime of infinite XUV pulse duration. Thus this regime where the lifetime of the state is longer than the driving pulse is not new. What is new is just how short they are able to go with the driving field duration.
3. “*dependence on τ implies that the minimum number of pump cycles N_{min} scales differently for different SBs with the important consequence that pulse duration alongside intensity can now be used to control the frequency components of the Floquet state.*”
 I would say that the pulse duration can be used to control the average population of the different Floquet bands, but not the instantaneous “frequency components of the Floquet state”

Manuscript ID: NCOMMS-22-18068A

Title: Controlling Floquet states on ultrashort time scales

Authors: M. Lucchini, et al.

List of changes:

- We carefully revised the manuscript and the supplementary information to avoid the use of the word population when it could lead the reader to think that we have measured an “instantaneous” Floquet population.
- We added a sentence to clarify that we do not use the pump-probe measurements to follow the system in real time, but we are interested in time integrated quantities.
- We changed the wording of the conclusion as suggested by the Referee to avoid possible misinterpretations.
- Section S1.2.1 has been largely rewritten and extended to better explain the reconstruction algorithm used and the origin of the effective IR intensity variation in order to reply to criticism #1.
- Following the Referee suggestion, we have changed the text related to her/his criticisms #2 and #3
- We added a new experimental reference (Ref. 38) to the manuscript.

We carefully considered all criticisms arose by the referee and modified the manuscript and the supplementary material to avoid any possible source of misinterpretation. In the following we give our extended point-by-point reply.

Reviewer #3

The authors have provided a detailed response and added some clarification which has greatly helped. In my opinion, the experiments and analyses are excellent (notwithstanding some concerns in the specific comment #1 below), however, I find their discussion of some of the implications misplaced.

We thank the Reviewer for recognizing the quality of our work. We have followed her/his suggestions and we do believe that the revised version of the manuscript frames better the discussion of the results and their implications.

The authors have clarified that the states being dressed by the IR laser pulse are free-electrons. As clarified, they are not free electrons, but the final state of the electrons in the continuum of states, still associated with the nucleus. Thus they are only present while the XUV laser pulse is present. This clarifies one of the previous confusions. This is, however, a very specific case of Floquet states, and in contrast to what is stated on line 53-54, I believe it does lose generality (and is not looking at “the simplest system: a free electron”).

To avoid any possible source of misinterpretation, we changed the sentence at lines 53-54 as follows:

“To find an experimental answer to these important questions without losing generality, we investigated the formation of a Floquet state of the simplest system capable of interacting with an external field: an electron in the continuum manifold of an atom – a quasi-free electron”

Given that a perfectly free electron cannot absorb/emit photons, **the simplest physical system capable of being dressed by a laser field and form a Floquet state is an electron in the continuum of an atom.**

Any other system, bound electron, electron in the continuum of a molecular system, etc., will require a more complex treatment.

Please note that the electrons we are studying are photoionized with considerable excess energy and pumped with a strong IR field. There are several theoretical and experimental works in literature (see our previous report for a more detailed discussion), proving that such an electron can be well treated with Volkov waves, i.e. the exact solution of a free electron in a field. Therefore, **our target is the best experimental realization of a free electron interacting with an external field.**

In this sense the referee is correct that this is not exactly a free-electron but rather a quasi-free electron. However, we maintain that this is the simplest system that one can study.

It is important to stress that whether or not the electron is bound or in the continuum of an atom is irrelevant to both the main message of our work and to its generality. The main goal of our work is to **investigate the limit of a theory strictly developed for monochromatic light when applied to pulses.** How many cycles do we need so that the system can be treated as periodic in time? In other words, how many oscillations of the pump field are needed to form a quasi-metastable Floquet state? This is a very fundamental question, which, to the best of our knowledge, has not been addressed so far to this level of detail. Knowing the necessary minimum number of pump cycles not only assures a correct interpretation of the experimental results, justifying the use of Floquet theory, but also sets the ultimate time resolution that can be achieved in pump-probe experiments which are based on this theory. Therefore we also maintain that our study is general because we investigate the generic conditions and timescales of the field dressing mechanism.

Many of the results and conclusion arise due to the interplay between the XUV and IR pulse durations. To be able to make general statements on the applicability of Floquet formalism for short pulses requires the effects of the XUV pulse duration to be taken into account. The authors do in fact do this through their detailed model and conclude that for driving pulses down to 2 cycles the results “can still be explained using an analytical model based on Floquet theory”. In other words, the reduction seen in the amplitude of the sidebands for short IR pulses is due to a combination of the IR and XUV pulse durations, and the Floquet formalism still holds for driving pulses as short as 10 optical cycles. This is an important point and valuable observation.

Where I disagree is with the assertion that under the specific conditions “~10 optical cycles are required to establish a Floquet ladder identical to the monochromatic case.” Certainly, under those conditions 10 optical cycles are required to generate side bands with the same amplitude as the monochromatic case, however, this is not the same as the time to establish the Floquet ladder, or as they claim, the “minimum number of driving cycles needed for a Floquet state $|\psi(t)\rangle$ to be established”.

Our observations and mathematical derivation prove that there is a strict connection between the photoelectron spectrum and the electron final state.

For the case of monochromatic pulses, our experimental results and the mathematical derivation presented in sections S2.1 to S2.4 of the supplementary proves that:

- 1) the final state of the electron is a Floquet (Volkov) state $|\psi(t)\rangle$
- 2) there is a precise relation between the photoelectron SB population and the amplitudes of the Floquet ladder states A_n .

These points are confirmed in literature (e.g. Ref. [17]) and are further proven to hold in our experimental conditions by the results reported in Fig. 2 of the manuscript.

For the case of finite (short) IR pulses, the mathematical derivation presented in the supplementary from sections S2.5 to S2.8 proves that if the finite IR envelope is properly considered, the photoelectron SB signal is again related to the amplitude of the Floquet ladder states A_n (Eq. (3) of the main manuscript). Furthermore, the final state can be considered as a Floquet-like state of the form:

$$|\psi'(t)\rangle = \sum_{n \neq 0} A_n g(t)^{|n|} e^{in\omega_0 t} |\alpha_n\rangle + [1 - (1 - A_0)g(t)^2] |\alpha_0\rangle \quad (\text{R1})$$

If the IR pulse is long enough for the approximation $g(t) \simeq 1$ to hold, the Floquet-like state $|\psi'(t)\rangle$ converges to a “pure” Floquet state

$$|\psi(t)\rangle = \sum_n A_n e^{in\omega_0 t} |\alpha_n\rangle \quad (\text{R2})$$

giving the same final photoelectron spectrum as the one found with monochromatic fields. **Therefore, for long enough IR pulses, the final dressed state created by the IR pulse is the same as the final state created by a monochromatic field and one will observe the same photoelectron spectrum, i.e. the same SBs.**

The full sentence in question reads:

*“Once the theoretical model has been validated, Eq. (4) can be used to estimate the minimum number of driving cycles needed for a Floquet state $|\psi(t)\rangle$ to be **established** as in the monochromatic case.”*

To be more accurate and avoid possible misinterpretations we have carefully revised the use of the word “establish” throughout the whole text. For example, the above sentence has been changed as follows:

*“Once the theoretical model has been validated, Eq. (4) can be used to estimate the minimum number of driving cycles needed for a Floquet state $|\psi(t)\rangle$ to be **observed** as in the monochromatic case.”*

The measured sideband amplitude is time-integrated, thus when the IR pulses become similar to or shorter than the XUV pulse, these amplitudes do not represent the instantaneous population or presence of the Floquet states. The asymptotic condition (Line 213) therefore does not represent the time to establish or populate the Floquet states. All of this is taken into account in the authors’ detailed and thorough model, which is followed down to 2 optical cycles and suggests that the instantaneous population of the Floquet states are established at least this fast. That is, they do not need 10 optical cycles to be established.

As it is important to clarify that we are **not** measuring instantaneous Floquet populations (see discussion below), we carefully revised the use of the word “population” in the main manuscript and in the supplementary, avoiding its use when not necessary. We also made sure to use the term sideband (SB) only when referring to the photoelectron spectrum in order to avoid confusion with the Floquet ladder states.

The authors’ detailed response in their rebuttal letter does not preclude these points. I agree with their detailed analysis, but again, not their conclusions. Fig R1 compares two different things (one the peak sideband amplitude, one effectively the integrated amplitude) and I do not disagree that their model is more complete than a simple phenomenological picture. The point remains, however, that the reduction in sideband amplitude is primarily due to the pulse overlap, and not any measured delay in formation or population of the Floquet states. If this is not the case, then I would welcome clarification, but as it stands I do not see any evidence of this.

- i) By no means we pretended to measure a delay in the “formation or population” of the Floquet states. We are quite puzzled by this comment as we carefully checked the manuscript, the supplementary and our previous point-by-point reply. **In none of the documents we mention that we measured/studied “a delay in formation/population” or “an instantaneous population” of the states.** We do not measure how the Floquet states are populated in real time during the interaction with each IR pulse of different duration. We do instead the following: we dress the target with shorter and shorter pulses. We see that if the pulses are long enough (10 cycles) the dressed state we create has the same properties of a Floquet state. Between 2 cycles and 10 cycles we found that the dressed state is not a Floquet state but can be interpreted as a Floquet-like state where the IR pulse envelope is considered. In this sense, we can state that a field-target interaction that lasts for more than 10 cycles is enough to observe a Floquet state that is equivalent to the one created by a monochromatic (CW) field. This statement is supported both by simulations and experimental results.
- ii) Eq. (S50) of the supplementary reports the Fourier transform in time of the scattering amplitude, related solely to the IR pulse (normalized by the XUV envelope). It does not contain the effect of the XUV pulse and yet shows a dependence on the IR pulse duration. Therefore, what we have observed is not only due to the IR-XUV overlap as suggested.

In Fig R2, I wouldn't expect varying the intensity and the pulse duration to give the same result, primarily because the XUV pulse duration relative to the IR pulse duration is also important. I agree with their point that "the Floquet sideband intensity can be changed by controlling the dressing pulse duration." However, what is claimed in the manuscript goes beyond just discussing the sideband intensity. As discussed above, the sideband intensity is a time-integrated measurements and is not the same as controlling the instantaneous population or presence of the Floquet states.

We would like here to stress that there is no problem/contradiction in observing the time-integrated photoelectron spectrum to search for the signature of Floquet states. This for two main reasons:

- i) Floquet states are periodic in time. They cannot be measured with one time snapshot, even if we had infinite time resolution. Their observation is to be found in the frequency domain as spatial periodic Bloch states are measured in momentum and not in real space over a single atomic site.
- ii) We do not need to resolve what happens during the interaction between the IR pulse and the target in real time. We rather want to compare together the "integrated" results of different pump-probe measurements.

It is worth noticing that full pump-probe scans are not strictly needed as in our analysis we evaluate the SB intensity at one specific delay: delay zero. Therefore, it would be sufficient for our analysis to measure only the photoelectron spectrum at $\tau = 0$ fs. We decided, instead, to acquire full pump-probe scans for each IR pulse so that we could improve the accuracy with which we determine the pump-probe overlap as explained in the supplementary. Furthermore, the acquisition of full pump-probe scans allows us to finely calibrate the experimental parameters on target (see our reply to the specific point #1).

Since it is important that the reader does not get the impression that we are trying to follow Floquet states in real time with our pump-probe scans, we added to the manuscript the following sentence:

"Complete pump-probe scans are here performed not with the intent to follow the SB population in real time during the laser-target interaction, but to precisely characterize the light pulses and locate the pump-probe temporal overlap where the signature of Floquet-like states is to be found."

Thus I continue to disagree with the claim that "the Floquet ladder population is strongly influenced also by the temporal profile of the exciting pulse". Again, however, if the authors are able to show evidence of a deviation of the instantaneous population of Floquet states from the Floquet formalism (incorporating the finite XUV and IR pulse durations) then this claim would be more reasonable.

- i) Floquet SBs can only be seen in integrated quantities. **We are not aware of any other way they have been observed experimentally.**
- ii) There is no "*instantaneous population or presence of the Floquet states*". Floquet states are periodic in time and cannot be populated instantaneously. This indeed is the exact reason why people do wonder if short pulses are able to induce Floquet states. Because if the field does not have enough oscillations (i.e. if it is not periodic enough) Floquet states cannot establish. If a Floquet state could be instantaneously populated there would be no need to look for a minimum pulse time duration.

Once clarified this, we do agree with the Referee that the sentences "*the Floquet sideband intensity can be changed by controlling the dressing pulse duration.*" and "*the Floquet ladder population is strongly influenced also by the temporal profile of the exciting pulse*" may be misleading as a strict Floquet state does not exist with IR pulses. To convey the message and avoid any possible source of misinterpretation /confusion, we changed the above sentences as follows:

"By demonstrating that the population of the amplitude and number of Floquet-like sidebands can be controlled not only with the driving laser pulse intensity and frequency..."

"Furthermore, since our study proves that the Floquet-like ladder that is established by short pulses is strongly influenced by the temporal profile of the exciting pulse..."

Further specific comments:

1. For the data in Fig 3c, the experimental spectra are corrected for the residual intensity fluctuations, the blue shift of the IR central wavelength and the deviations of the pulse envelopes from an ideal Gaussian shape (see Supplementary Section S1.2.1), The raw data are shown in FigS2 and show significantly more scatter. It seems from my understanding that the process for correcting these relies on fitting the delay dependent photoelectron spectra. One of the largest corrections seems to be to the IR pulse intensity, which varies by over a factor of 2. It is not clear to me why this is necessary when the pulse properties are determined by FROG and the intensity is measured based on these. What is the reason for these differences? What is perhaps more concerning is that in the fitting process, the sidebands are used and there is a term in the equation describing them goes as the electric field amplitude of the XUV beam multiplied by the amplitude of the IR beam, just as there is in Eq4 of this manuscript. While I haven't fully delved into the fitting procedure, I am a little worried that there may be a circular argument here, where the expected response is used to determine the intensity (ie the fitting procedure detailed in S1.2 and Ref S35), which is then used to shift the experimental data points in Fig 3c, which then agree with the model (Eq4) and is used as evidence that the model is accurate and followed by the data. I acknowledge that the models are different, but there are certainly similarities, and it would aid clarity if the authors can give more detail on this fitting. It would also help if they give some explanation as to why the IR intensity varies by more than a factor of two from what was expected.

We thank the Referee for this comment as it allows us to clarify a relevant aspect of our data analysis. To reconstruct the properties of IR and XUV pulses we did not simply perform a fit of the experimental data. We removed the word “fit” and the reference to Moio et al. from the related paragraph as, without a more detailed explanation, they appear to be misleading. Rather than applying a fit we adopted an iterative algorithm based on the FROG approach, modified to explicitly consider all the parameters that are known from independent measurements with high enough accuracy, namely the IR pulse temporal profile, the XUV spectral intensity and the TOF instrumental response. To avoid any misinterpretation, be more accurate and exhaustive, we have rewritten the related section of the supplementary.

In brief, we applied an iterative algorithm that follows a FROG-like approach. Starting from an educated guess for XUV and IR pulses the resulting differential spectrogram is calculated using the SFA formula (Eq. (5) of the manuscript)

$$\Delta S'(\omega, \tau) = \left| \int_{-\infty}^{\infty} E_x(t - \tau) e^{-i \int_t^{\infty} \frac{1}{2}(p+A_{IR}(t'))^2 dt'} e^{iI_p t} dt \right|^2 - |\hat{E}_{X0}(\omega)|^2$$

and including the instrumental response function which has been independently characterized. Corrections to the XUV and IR pulses are calculated starting from the difference between $\Delta S'(\omega, \tau)$ and the measured differential spectrogram $\Delta S(\omega, \tau)$. Before using the updated XUV and IR functions to start the next iteration, the code imposes the IR temporal envelope to coincide (within the experimental accuracy) to the one independently measured by a second-harmonic FROG and the XUV spectral intensity to the measured harmonic spectrum. To speed the procedure, we use a Taylor time expansion of the complex exponential in the integral instead of using a Fourier expansion as in the Floquet-like model. When the distance between $\Delta S'(\omega, \tau)$ and $\Delta S(\omega, \tau)$ reaches a threshold value, the algorithm stops and delivers the reconstructed IR and XUV pulses.

The model used in the reconstruction is based on Eq. (5) and not on Eq. (3) (the Referee suggests Eq. (4) which must be a typo), introducing a global central momentum approximation (CMA) instead of a local CMA as in Eq. (3). Therefore, we are confident that we are not applying a “circular argument”. Please note that after calibration the experimental data points do not perfectly agree with the model. This would be instead the case if a circular argument were adopted.

It is further important to note that:

- i) the simplified model is first validated against numerical simulations performed with Eq. (5), without additional approximations. Its validity thus does not solely depend on the experimental results.

- ii) the qualitative behaviour predicted by the model is already observed in the raw, uncalibrated, data. Therefore, both our theoretical analysis and the main experimental finding (Floquet-like SBs of lower intensities are observed with short IR pulses) are robust.

In principle, during the reconstruction it is also possible to fix the IR intensity to match the one estimated in the lab. We decided not to proceed in this way as the intensity estimation is less reliable. To estimate the intensity, we measure the IR average power with a power meter, its temporal profile with a FROG and its spatial profile with a beam profiler. While the first two are accurate enough, the measurement of the spatial profile is less accurate for two reasons: 1) the detection system suffers of an important numerical background whose accurate removal is far from being trivial. According to the background threshold used the estimated IR intensity can easily change by a relevant amount; 2) the effective intensity felt by the atoms depends on the degree of spatial overlap between IR and XUV beams at the target. Any geometrical misalignment will therefore cause a different effective IR intensity. These two effects together explain the discrepancy between the estimated and reconstruction IR intensity. We have added this explanation in section S1.2.1.

2. The authors state: Most interestingly, our results show that Floquet-like bands can be observed, albeit with a reduced amplitude, also if the XUV (probe) pulse is longer than the IR (pump), $\sigma_X > \sigma_{IR}$, therefore extending the previously known regime of applicability of Floquet theory. Floquet theory is primarily used to describe dressing of equilibrium states, which equivalent to a regime of infinite XUV pulse duration. Thus this regime where the lifetime of the state is longer than the driving pulse is not new. What is new is just how short they are able to go with the driving field duration.

While we do agree that Floquet theory is primarily used to describe dressing of equilibrium states, we do not fully agree that this is equivalent to a regime of infinite XUV pulse duration. In our experiment the IR behaves as a pump, dressing the atomic states in the continuum, i.e. creating the Floquet state even before the transition mediated by the XUV happens. The XUV excites the electron from the initial state to the Floquet state, probing its existence. The XUV duration does not affect how the IR dresses the final state, but rather how this final state is probed. So we do not see the direct link between the time duration of the probe and the properties of the Floquet state induced by the pump.

This said, Ref. 14 of the main manuscript states that clear Floquet sidebands are observed only if $\sigma_{pump} > \sigma_{probe} \gg T_{pump}$, which for us means $\sigma_{IR} > \sigma_X \gg T_{IR}$. This time hierarchy is commonly used as a criterion to identify the correct regime in pump-probe experiments with Floquet states (see for example a more recent work: Lukas Broers and Ludwig Mathey Phys. Rev. Research **4**, 013057 (2022), <https://doi.org/10.1103/PhysRevResearch.4.013057>, now added as Ref. 38). As in Fig. 4a of the main manuscript $\sigma_X \approx 4.12T_{IR}$, the simulations results with $2T_{IR} < \sigma_{IR} < 4T_{IR}$ show that Floquet-like SBs, can be observed also if $\sigma_{pump} < \sigma_{probe}$, albeit with a reduced amplitude. This condition, $\sigma_{pump} < \sigma_{probe}$, is also matched for the experimental point obtained with IR duration of 9 fs (the experimental XUV lasts for about 12 fs) and reported in Fig. 3c. To the best of our knowledge, it is the first time that this regime has been experimentally investigated and interpreted within a Floquet-like picture.

Finally, please notice that Fig. 3b of the main manuscript shows that the SBs obtained with IR pulses of finite duration exhibit a decreasing amplitude with increasing XUV duration. The monochromatic limit for the SB amplitude is not reached for monochromatic XUV pulses, but rather the opposite. It is therefore important to stress that Floquet-like sidebands can be found also when $\sigma_{pump} < \sigma_{probe}$.

To better clarify our message, we have changed the sentence appointed by the Referee which now reads as follows:

“Conversely from what suggested in the recent literature^{14,38}, our results show that Floquet-like bands can be observed, albeit with a reduced amplitude, also if the XUV (probe) pulse is longer than the IR (pump), $\sigma_X > \sigma_{IR}$, therefore proving that the Floquet theory can be extended to interpret the experimental results even if the hierarchy $\sigma_{IR} > \sigma_X \gg T_{IR}$ is not strictly matched.”

3. *“dependence on $|n|$ implies that the minimum number of pump cycles N_{IR} scales differently for different SBs with the important consequence that pulse duration alongside intensity can now be used to control the frequency components of the Floquet state.” I would say that the pulse duration can be used to control the average population of the different Floquet bands, but not the instantaneous “frequency components of the Floquet state”*

As we now clarified, we do not pretend to follow the **instantaneous** frequency components of the Floquet state. To do that we would need to define a sort of instantaneous population of the Floquet ladder in time (that we believe to have no strict physical meaning) and perform a time-gated Fourier transform (a sort of Wigner or Gabor analysis) as a simple Fourier transform is not time-resolved by definition. When we write “frequency components” they have to be intended as the amplitudes of the monochromatic components which define the signal in time (standard definition). Certainly, they are closer to an average property of the system rather than an instantaneous one.

As nowhere in our work we mention a time-gated kind of analysis, we do not fully understand where this doubt came from. Anyway, to be rigorous and avoid misinterpretations we changed the sentence as follows:

“Finally, the explicit dependence on $|n|$ implies that the minimum number of pump cycles N_{IR} scales differently for different SBs with the important consequence that pulse duration alongside intensity can now be used to control the average population of the Floquet-like bands.”

REVIEWERS' COMMENTS

Reviewer #3 (Remarks to the Author):

The authors have once again provided a detailed response to the issues raised, and while I still disagree on some aspects, I think the changes made to the manuscript ameliorate the main issues. In particular, I appreciate the additional text describing how Floquet theory can be used for pulses shorter than 10fs, and shorter than the probe, which I believe is an important point in considering the limits of Floquet theory for short pulses.

There are several aspects in their response that we could continue discussing for some time, and indeed, I agree with >80% of what they say. I would however make the following points that the authors might consider.

In their response the authors state that “*Floquet SBs can only be seen in integrated quantities. We are not aware of any other way they have been observed experimentally*”. I agree that my previous use of the term instantaneous was not accurate. However, the point I was attempting to make was that there is still some temporal evolution of the Floquet states as the driving field turns on and off. The question of measuring this switching on process is indeed related to the question of how short the driving pulse can be – that is, how many optical cycles are required to be able to establish (pump) and measure (probe) the Floquet states. For the case of probe pulses shorter than the pump pulse, it is possible to obtain some dynamics, as has been done in past pump-probe measurements (eg Sie et al, Nature Materials 14, 290 (2015)). Measuring such dynamics has the potential to resolve deviations from predicted Floquet behaviour, although the effects of the duration of the probe will also be convolved here.

The authors state in their reply: “*Between 2 cycles and 10 cycles we found that the dressed state is not a Floquet state but can be interpreted as a Floquet-like state where the IR pulse envelope is considered. In this sense, we can state that a field-target interaction that lasts for more than 10 cycles is enough to observe a Floquet state that is equivalent to the one created by a monochromatic (CW) field.*”. **It is this regime where we continue to differ in how this should be described. I would once again suggest that there is no clear evidence that the dressed state in this regime is not a Floquet state. The major reason that IR pulse envelope is important is because it is now shorter than the XUV pulse. In the manner described by the authors in their most recent response:** “*the IR behaves as a pump, dressing the atomic states in the continuum, i.e. creating the Floquet state even before the transition mediated by the XUV happens. The XUV excites the electron from the initial state to the Floquet state, probing its existence. The XUV duration does not affect how the IR dresses the final state, but rather how this final state is probed.*” **Following this reasoning, when the IR beam is shorter than the XUV beam, fewer electrons created by the XUV beam are able to probe the Floquet states and hence the sidebands have reduced amplitude. This is independent of any question of whether or not the IR beam is able to fully establish the Floquet states. To provide a convincing argument that the IR pulse duration causes an additional effect beyond this, deviations of the data from a model where only this effect is considered should be analysed.**

Regarding Eq. (S50), as I understand this removes the effects of the XUV pulse, effectively leading to a constant total electron population (as stated, it gives “*the time evolution of the populated final Floquet state*”). This is again the case where the probe is the shortest pulse duration, and the number of electrons that can be scattered into the Floquet sideband is determined by the duration of the IR beam (pump). I do not see that this shows any deviation from Floquet behaviour.

Manuscript ID: NCOMMS-22-18068B

Title: Controlling Floquet states on ultrashort time scales

Authors: M. Lucchini, et al.

List of changes:

- In reply to one of the Referee's comments and underline the fact that the XUV is not always longer than the IR pulse, we added a sentence in the discussion of the minimal number of IR cycles needed to clarify the difference between the optimal condition and the experimental reported case.
- To further stress the above point, we introduced a shaded area in Fig. 4a showing the region where the IR duration is shorter than the XUV duration. The figure caption has been changed accordingly.
- We added a sentence in the main text, before Fig. 4, stating that the results of Fig. 4b already show that the observed behaviour cannot be simply described by a reduced XUV probing capability, as suggested by the Referee.
- We corrected two sentences, one in the main text and one in the supplementary, to clarify the meaning of the quantity $|s(t)|^2$.
- A new figure (now figure S9) has been added to the Supplementary to visualize how the prediction of Eq. (S50) deviates from the Floquet case (Eq. (S29)).
- We applied all the changes requested by the authors checklist file.

We carefully considered all criticisms arose by the Referee and modified the manuscript and the supplementary material. In the following we give our extended point-by-point reply.

Reviewer #3

The authors have once again provided a detailed response to the issues raised, and while I still disagree on some aspects, I think the changes made to the manuscript ameliorate the main issues. In particular, I appreciate the additional text describing how Floquet theory can be used for pulses shorter than 10fs, and shorter than the probe, which I believe is an important point in considering the limits of Floquet theory for short pulses. There are several aspects in their response that we could continue discussing for some time, and indeed, I agree with >80% of what they say. I would however make the following points that the authors might consider.

We thank the Reviewer for the positive comment. We are confident that our latest revised version matches also the remaining 20%.

In their response the authors state that "Floquet SBs can only be seen in integrated quantities. We are not aware of any other way they have been observed experimentally". I agree that my previous use of the term instantaneous was not accurate. However, the point I was attempting to make was that there is still some temporal evolution of the Floquet states as the driving field turns on and off. The question of measuring this switching on process is indeed related to the question of how short the driving pulse can be – that is, how many optical cycles are required to be able to establish (pump) and measure (probe) the Floquet states. For the case of probe pulses shorter than the pump pulse, it is possible to obtain some dynamics, as has been done in past pump-probe measurements (eg Sie et al, Nature Materials 14, 290 (2015)). Measuring such dynamics has the potential to resolve deviations from predicted Floquet behaviour, although the effects of the duration of the probe will also be convolved here.

We thank the Referee for the clarification. It seems that he/she was interested in what happens in time during the pulse leading and trailing edges. While we recognize this to be an interesting topic, we plan to investigate this aspect in the future. At present we aim at investigating how to treat the response of shorter and shorter pulses and not to resolve what happens within the pulse, therefore the direction proposed by the Referee falls outside the scope of our work.

The authors state in their reply: “Between 2 cycles and 10 cycles we found that the dressed state is not a Floquet state but can be interpreted as a Floquet-like state where the IR pulse envelope is considered. In this sense, we can state that a field-target interaction that lasts for more than 10 cycles is enough to observe a Floquet state that is equivalent to the one created by a monochromatic (CW) field.” It is this regime where we continue to differ in how this should be described. I would once again suggest that there is no clear evidence that the dressed state in this regime is not a Floquet state.

Strictly speaking, within Floquet theory, the term Floquet state indicates a dressed state, perfectly periodic in time, obtained with a perfectly periodic driving. Therefore, regardless the duration of the pulse, the use of the term “Floquet state” is by definition not legitimate. The final state can be still considered a good approximation of a Floquet state only if the driving pulse is long enough to be a good approximation of a monochromatic pulse. This is certainly not the case for a pulse with a number of cycles between 2 and 10.

In our view, the problem is exactly the opposite to the one proposed by the Referee. Rather than searching for a “clear evidence that the dressed state in this regime is not a Floquet state”, one should provide evidence that the state can nevertheless be considered a Floquet state, since, by definition, it is not.

The major reason that IR pulse envelope is important is because it is now shorter than the XUV pulse. In the manner described by the authors in their most recent response: “the IR behaves as a pump, dressing the atomic states in the continuum, i.e. creating the Floquet state even before the transition mediated by the XUV happens. The XUV excites the electron from the initial state to the Floquet state, probing its existence. The XUV duration does not affect how the IR dresses the final state, but rather how this final state is probed.” Following this reasoning, when the IR beam is shorter than the XUV beam, fewer electrons created by the XUV beam are able to probe the Floquet states and hence the sidebands have reduced amplitude.

We would like first to clear out an important possible misunderstanding: the IR pulse is not always shorter than the XUV in our paper. The analysis of Fig. 4c of the main manuscript is performed for different durations of the XUV pulse, expressed in units of the IR cycle. The number of IR cycles needed to converge increases with the duration of the XUV and with the order of the SBs involved. In the related discussion we have analysed the best condition, i.e. the condition for which convergence is reached earlier. This condition is matched with an XUV that lasts only 2 IR cycles. In the manuscript we write: “If we consider the shortest XUV pulse for which clear SBs are observed, i.e. $N_X = 2, \dots$ ”. For this XUV duration, we found that if the IR lasts between 2 and 10 cycles, the dressed state is not a Floquet state, but can be interpreted as a Floquet-like state where the IR pulse envelope is considered. Above 10 cycles the pulse is a good approximation of a monochromatic driving, i.e. we have a “pure” Floquet state. Below 2 cycles our model fails as reported in Fig. 4a. To underline this even better, we have now added a vertical line in Fig. 4a marking the case for which IR and XUV have the same duration.

The situation is different in Figs. 2 and 3, where the XUV pulse lasts for ~ 12 fs, i.e. almost 4.5 IR cycles. Using the equation reported in the text, $N_{IR} = \frac{(1-\alpha)\sqrt{|n|}}{\sqrt{2\alpha-\alpha^2}} N_x$, we can calculate that 3τ -convergence is reached in 13.7 cycles for $n = 1$ and 19.4 cycles for $n = 2$. To clarify this point, we added the following sentence in the discussion of Fig. 4:

“This is different from the experimental condition used in Fig. 3 where the XUV pulse lasts almost 4.5 cycles. In that case 3τ -convergence is reached after $19.4T_{IR}$ for $n = \pm 2$ and $13.7T_{IR}$ for $n = \pm 1$.”

This is independent of any question of whether or not the IR beam is able to fully establish the Floquet states. To provide a convincing argument that the IR pulse duration causes an additional effect beyond this, deviations of the data from a model where only this effect is considered should be analysed.

We would like to stress that our results of Fig. 3 are plotted against the IR duration, normalized by the XUV duration. The effect of the finite XUV duration is therefore accounted for. Furthermore, a prove that the observed variations do not originate only from the XUV being less efficient in probing the final state created by the IR is already given in Fig. 4b. In this figure the XUV duration, σ_x , is always lower than the IR duration, $\sigma_{IR} = 10T_{IR}$. If the observed effect were only due to a reduced time overlap between the XUV and the IR (as suggested by the Referee), there should be not time dependence below $\sigma_x = 10T_{IR}$. Nevertheless, the SB signal keeps increasing with decreasing σ_x , further proving that what we observe goes beyond a simple overlap effect. To emphasize this aspect, we have added the following sentence to the discussion of Fig. 4b:

“The non-trivial behaviour of the SB amplitudes in Fig. 4b, where the XUV pulse is always shorter than the IR pulse, shows that the observed dependence cannot be solely attributed to a reduced probing temporal overlap between the IR and XUV pulses.”

Regarding Eq. (S50), as I understand this removes the effects of the XUV pulse, effectively leading to a constant total electron population (as stated, it gives “the time evolution of the populated final Floquet state”). This is again the case where the probe is the shortest pulse duration, and the number of electrons that can be scattered into the Floquet sideband is determined by the duration of the IR beam (pump). I do not see that this shows any deviation from Floquet behaviour.

We thank the Referee for this comment Eqs. (S29) and (S50) represent the same physical quantity: “the time evolution of the scattering amplitude without considering the XUV temporal properties”. Eq. (S29) is derived for a monochromatic IR, i.e. for Floquet final states, while Eq. (S50) is derived for IR pulses, i.e. for Floquet-like states. In the previous revision, we corrected the description for Eq. (S29) removing any possible association to the fuzzy concept of Floquet population. We have now also corrected the description of Eq. (S50) which reads:

“In this limit, the time evolution of the scattering amplitude without considering the XUV temporal properties goes as:

$$|s(t)|^2 \propto \left| \sum_{n \neq 0}^{\infty} A_n(p_n, E_0, \omega_0) g(t)^{|n|} e^{in\omega_0 t} + A'_0(t) \right|^2 \quad (S50)''$$

To be consistent and avoid possible misinterpretation, we also revised the related sentence in the description of Fig. 2c:

“...to become visible in the time evolution of scattering amplitude related to the IR pump only, $|s(t)|^2$...”

While we agree that Eq. (S50) “removes the effects of the XUV pulse”, it does not lead to a “constant total electron population” as stated by the Referee. As it does not contain the XUV effect, it does not fully describe the ionization, so it cannot describe the electron population. Instead, it describes how the time behaviour of the scattering amplitude is affected solely by the IR. For the case of monochromatic pulses (insets of Fig. 2c), Eq. (S50) becomes Eq. (S29). It does not give a constant output, but rather a perfectly periodic quantity (signature of a Floquet state). For the case of IR pulses, Eq. (S50) predicts a dependence also on the IR pulse envelope (signature that this cannot be considered a Floquet state).

Both Eqs. (S29) and (S50) do not depend on the XUV pulse properties, so we do not understand why the Referee is convinced that “This is again the case where the probe is the shortest pulse duration, ...”. It is not. While deriving those equations, the only hypothesis made on the probe side is that the XUV bandwidth should not be wider than roughly 2 IR photons to assure negligible spectral overlap between the different spectral components (see the discussion below Eq. (S22) in the supplementary). The XUV can be longer than the IR, the two equations will not change.

For what concerning the last comment “I do not see that this shows any deviation from Floquet behaviour”, the fact that Eq. (S50) is not the same as Eq. (S29) it’s itself evidence. Indeed, any difference between what

predicted by Eq. (S29), describing Floquet states, and what predicted by Eq. (S50), derived by IR pulses, stems exactly from a deviation from the Floquet behaviour.

Figure R 1: **(a)**, time evolution of scattering amplitude related to the IR pump only, $|s(t)|^2$, calculated for an IR intensity of $5 \times 10^{11} \text{ W/cm}^2$. The grey curve represents the monochromatic case (Floquet state, Eq. (S29)), while the coloured curves show the prediction of Eq. (S50) for σ_{IR} equal: 146 fs, dashed-blue, 43.3 fs ciano, 21.1 fs green and 9.2 fs red. These conditions correspond to the experimental points in Figs. 2c and 3c of the main manuscript. **(b)**, Deviations of the pulsed case from the monochromatic scenario. Same colour coding as in (a).

Figure R1a shows the behaviour of $|s(t)|^2$ calculated for a Floquet state (light grey, Eq. (S29)) and for driving pulses (Eq. (S50)) of different durations σ_{IR} : 146 fs, dashed-blue, 43.3 fs ciano, 21.1 fs green and 9.2 fs red. The IR intensity used is $5 \times 10^{11} \text{ W/cm}^2$. Hence the curves coincide with the second inset of Fig. 2c, grey curve, and the four insets of Fig. 3c, coloured curves. Figure R1b shows the difference between $|s(t)|^2$ of a Floquet state and the one induced by a pulsed driving (same colour coding as in Fig. R1a), visually representing why Eq. (S50) deviates from the Floquet behaviour of Eq. (S29).

For the sake of cleanness, Fig. R1 has been added to the Supplementary at the end of section S2.6.